# Wheel Tread Reconstruction Based on Improved Stoilov Algorithm

Tao Tang , Jianping Peng *, Jinlong Li , Yingying Wan, Xingzi Liu and Ruyu Ma

Institute of Optoelectronic Engineering, College of Physical Science and Technology, Southwest Jiaotong University, Chengdu 610036, China; tangtao_swjtu@126.com (T.T.); jinlong_lee@126.com (J.L.); yingyingwan@swjtu.edu.cn (Y.W.); xingzi_swjtu@126.com (X.L.); maruyu97@126.com (R.M.)
* Correspondence: peng.jian.ping@126.com

**Abstract:** With the development of rail transit in terms of speed and carrying capacity, train safety problems caused by wheel tread defects and wear have become more prominent. The wheel is an important part of the train, and the wear and defects of the wheel tread are directly related to the safety of the train; therefore, wheel tread testing is a key element of train testing. In phase measuring profilometry (PMP), the virtual sine grating generated by the computer is projected onto the measured wheel tread by a digital projector, and then a camera is used to obtain the modulated deformed grating on the surface of the wheel tread. Next, the wrapped phase is obtained by the improved Stoilov algorithm, and the unwrapped phase is obtained by the phase unwrapped algorithm. Finally, the three-dimensional (3D) profile of the wheel tread is reconstructed. This paper presents an improved Stoilov algorithm based on probability and statistics. Supposing that the probability of real data was the highest, we chose the cosine square matrix value of the phase shift for processing. After ruling out the singular points of large error, we obtained the closest value to the true phase shift using the method of probability and statistics. The experimental results show that this method can effectively restrain the singular phenomenon, and the 3D profile of wheel tread can be reconstructed successfully.

**Keywords:** phase measuring profilometry; Stoilov algorithm; wheel tread; reconstruction

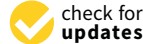



## 1. Introduction

The wheel is an extremely important element ensuring the safe operation of a train, which plays a load-bearing and guiding role. Once there is something wrong with the wheel, the wheel will collapse and cut the shaft, which will further lead to serious accidents. Wheel wear and surface defects have a great effect on the safety, smoothness, running speed, and passenger comfort of railway operation. Therefore, it is very important to detect the wheel status and ensure safety [1,2] by detecting the shape of the wheel tread. There are two commonly used methods for wheel tread detection; one is the contact-type detection method, while the other is the noncontact detection method [3,4]. Manual detection is the most commonly used contact method for traditional wheel detection, but it has some shortcomings, such as incurring a high cost, being labor-intensive and time-consuming, the inability to observe directly, and the possibility of missed detection. The noncontact detection method combines the integration of light, machine, and electricity, which has attracted interest due to its noncontact nature, high precision, fast speed, small environmental impact, and good real-time processing ability. It effectively addresses the difficulties in evaluating wheel tread defects with accurate measurements [5–8]. Its applications range from measuring the 3D shape of microelectromechanical system (MEMS) components to measuring the flatness of large panels. This technique has found various applications in diverse fields: biomedical applications such as 3D intraoral dental measurements, noninvasive 3D imaging and monitoring of vascular wall deformations, industrial and scientific applications such as characterization of MEMS components, measurement of surface roughness, reverse engineering, quality control of printed circuit board manufacturing, and

heat-flow visualization [9,10]. In recent years, we have been committed to the research of digital fringe projection and phase shift technology; we developed a variety of algorithms to obtain the phase, and we used programming techniques to improve the processing speed [11–14].

In 3D measurement, phase shift technology can be divided into fixed-step phase shift algorithms and equal-step phase shift algorithms [15,16]. In recent years, the Stoilov algorithm in the application of phase shift interferometry only requires that the phase shift step is equal, and there is no need to determine the step value and the total phase shift of $2\pi$ integer multiples. It is widely used in phase shift interferometry. The Stoilov algorithm was introduced into the phase measuring profilometry (PMP), which makes the phase measurement profilometry more flexible in measuring the object. However, due to the influence of ambient light, system digitization error, and detector nonlinearity, the calculation method of phase shift in the traditional S algorithm may lead to a large error, and which is propagated during phase unwrapping, resulting in a larger area of dephasing error, which finally affects the accuracy of phase extraction and system measurement [17–19].

Srinivasan and Haliouas et al. [20–22]. proposed phase shift interferometry (PSI) for the first time in the early 1980s and used this technique to measure the three-dimensional surface of an object, but its accuracy needed to be improved. This method is called phase measurement profilometry (PMP). Stoilov et al. (1997) [17] proposed a new formula for calculating the phase in interferometric measurements using the phase-stepping method, but they only conducted a simulation and did not measure an object. Xingfen et al. [23] proposed a new method to improve the Stoilov algorithm using a statistical approach. In this method, the phase error brought by the abnormal phenomena is restrained, and the precision of the 3D measurement is improved, but the outcome did not reach the level of satisfaction. Yingchun et al. [24] proposed a new method to actively control the phase shift to improve the Stoilov algorithm, avoiding the influence of ambient light, system digitization error, and detector nonlinearity on the phase shift calculation, and improving the accuracy of phase extraction and system measurement. However, as with traditional phase measurement profilometry, it is necessary to obtain accurate phase shift. Yiping et al. [25] proposed a method of grating parameter optimization using the statistical approach Stoilov algorithm, effectively improving the accuracy, but leaving room for further improvement. Ting et al. [26] proposed a new method to improve the Stoilov algorithm according to short-distance priority and weighted mean, effectively removing impulse noise, which may have an advantage in protecting phase details. However, it is easily affected by ambient light, and its robustness needs to be further improved. This paper presents a new method to improve the Stoilov algorithm using probability and statistics, effectively eliminating the singular phenomenon of reconstruction on the wheel tread.

## 2. Materials and Methods

The principle of PMP, as shown in Figure 1, is to project standard grating fringes onto the measured object for spatial phase modulation, and then capture five deformed grating fringe patterns with a camera. The distance between the digital light-processing (DLP) pupil center and the camera pupil center is defined as L. The distance between the center of the DLP pupil and the reference plane is d. Due to the height of the inspection point Q, the fringe is deformed from the $x_o$ point to $x_r$ point; thus, $(x_o - x_r)$ is the displacement of the stripe in world coordinates (X–Y). Geometrically, the height of the measuring point z relative to the reference point can be expressed as the geometry of the digital fringe projection (DFP) experimental device and as a function of the scalars $x_r$ and $x_o$, as shown in Figure 1 [27].

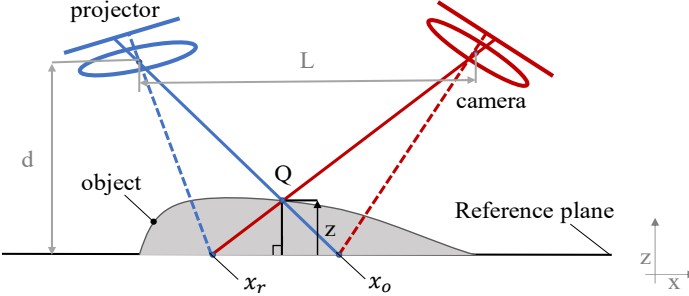

**Figure 1.** The principle of PMP.

However, the Stoilov algorithm is a five-step phase shift algorithm, requiring five deformed fringes. The five deformed patterns $I_n(x,y)$, $(n = 1, 2, 3, 4, 5)$ are shown in Equation (1).

$$I_n(x,y) = R(x,y)\{A(x,y) + B(x,y)cos[\varnothing(x,y) + (n-1)\varphi_0]\}(n = 1,2,3,4,5), \quad (1)$$

where $R(x,y)$ is the surface reflectivity of the object, $A(x,y)$ is the background light intensity, $B(x,y)$ is the fringe contrast, $\varnothing(x,y)$ indicates phase modulation factor including the height information of the object, and $\varphi_0$ is the equivalent phase shift caused by the equivalent step shift; $\varphi_0$ is a constant. According to the Stoilov algorithm [17], phase distribution can be calculated using Equation (2).

$$\varnothing(x,y) = arctan\left\{\frac{2[I_2(x,y) - I_4(x,y)]}{2I_3(x,y) - I_1(x,y) - I_5(x,y)} sin\varphi_0\right\}, \quad (2)$$

In this equation,

$$sin\varphi_0 = \sqrt{1 - \left\{\frac{I_1(x,y) - I_5(x,y)}{2[I_2(x,y) - I_4(x,y)]}\right\}^2}. \quad (3)$$

An arctangent function is used so that the phase value obtained from Equation (2) ranges from $-\pi$ to $+\pi$ with a $2\pi$ modulus. Spatial or temporal phase unwrapping algorithms can be used to unwrap the phase so that we can get a continuous phase map. The unwrapping process essentially estimates the $2\pi$ discontinuous locations and removes the $2\pi$ jumps by adding or subtracting $k(x,y)$ multiples of $2\pi$. The continuous phase map $\Psi(x,y)$ can be obtained by applying a phase unwrapping algorithm to determine the fringe order $k(x,y)$.

$$\Psi(x,y) = \varnothing(x,y) + k(x,y) \times 2\pi. \quad (4)$$

The unwrapped phase can be used for 3D reconstruction once the system is calibrated. The height information of the measurement object $h(x,y)$ can be reconstructed by the mapping algorithm [28,29].

$$\frac{1}{h(x,y)} = a(x,y) + b(x,y)\frac{1}{\Psi(x,y)} + c(x,y)\frac{1}{\Psi^2(x,y)}. \quad (5)$$

In Equation (5), $a(x,y)$, $b(x,y)$, and $c(x,y)$ are obtained by calibration.

From Equation (2), we can know that the calculation accuracy of $sin\varphi_0$ in Equation (3) influences the phase extraction. Due to the digitization error of the system, the nonlinearity of the detector, and the influence of ambient light, five deformed fringes appear as errors, and the following anomalies occur:

(1) In the first case, where $I_2 = I_4$, the denominator is 0, and when $sin\varphi_0$ is meaningless, the value of $sin\varphi_0$ depends on the light intensity of the captured image. When the light intensity $I_2 = I_4$ collected by a camera has an error and is presented as an integer

gray value, resulting in $I_2 = I_4$ in some pixel positions, sin is meaningless, which affects the unwrapping phase.

(2)     In the second case, the square operation leads to the plural situation of $sin\varphi_0$, and if the surface of the object is not smooth, there is a sudden height jump, which leads to $\left\{ \frac{I_1(x,y)-I_5(x,y)}{2[I_2(x,y)-I_4(x,y)]} \right\}^2 > 1$, resulting in complex phase and error.

(3)     In the third case, when the cosine value of the phase shift is not equal to zero, but $I_1 = I_5$ at some points, this makes the phase shift identically equal to $\pi/2$, resulting in phase shift calculation error.

(4)     In the fourth case, when the denominator $2(I_2 - I_4)$ is very small, tiny changes of the numerator $I_1 - I_5$ lead to a large error in $sin\varphi_0$.

The differences between the method we used and the maximum likelihood (ML) estimation are as follows:

(1)     Maximum likelihood estimation needs to know the wrapped phase $\varnothing(x,y)$ of the unwrapped phase and then estimate the phase shift $\varphi_0$.

(2)     The unwrapped phase $\varnothing(x,y)$ is obtained through the improved Stoilov algorithm.

This process is irreversible; hence, we do not know the distribution of the unwrapped phase $\varnothing(x,y)$.

When using the Stoilov algorithm to calculate $sin\varphi_0$, although there are four kinds of anomalies as mentioned above, most of the numbers are close to the true value. If the singularities of the above four characteristics can be gradually excluded from the $sin\varphi_0$ distribution, and the remaining numbers close to the true values are statistically approximated, i.e., the average values of these numbers are used as realistic values to compensate for the values of each point in the $sin\varphi_0$ distribution, the dephasing error caused by the above four abnormal conditions can be effectively avoided, and the influence of ambient light noise can be restrained to a certain extent. Let us assume the data of $SD = \left\{ (I_1 - I_5)/2[I_2 - I_4] \right\}^2$. The specific algorithm is as follows:

(1)     For the SD matrix, calculate $\Delta = I_2 - I_4$; if $\Delta = 0$, mark these points as 200.

(2)     Calculate $SD = \left\{ (I_1 - I_5)/2[I_2 - I_4] \right\}^2$; if $SD > 1$, mark them as 400.

(3)     The data of the SD matrix are stored in a one-dimensional array of N.

(4)     By searching the array N, find the singularities labeled 200 and 400, and exclude them from N. Set the remaining data to SA.

(5)     Sort the data in the SA array from small to large, whereby there are only two singularities left. Among all the data, the probability of the normal point is the highest.

A very large error should be greater or less than the average. That is to say, if the data are ranked, there will be several errors on both sides. Therefore, after sorting the SA array, we selected the middle 40% of the data to calculate the average. Using the above improved Stoilov algorithm, the errors caused by four kinds of singularities could be effectively eliminated. When there is a large error, it should be far away from the average. If we arrange the array SA from small to large, a large part of the error should exist at both ends of the array. We selected the middle 40% of the array to calculate its average. Through the above-described methods, the Stoilov algorithm could be improved, and the error caused by the reconstruction of four singularities could be effectively suppressed.

## 3. Results

To verify the performance of the new method, an experimental setup was developed (Figure 2), which mainly included a DLP projector (Light-Crafter 4500) with $912 \times 1140$ resolution and a complementary metal–oxide–semiconductor (CMOS) industrial black-and-white camera (Basler acA1920-40 gm) with $1920 \times 1200$ resolution and a 8 mm focal length lens. Moreover, the system also included a slide rail (MGW12H4R1200ZFCM) with an accuracy of 0.03 mm and a computer for calculation and control.

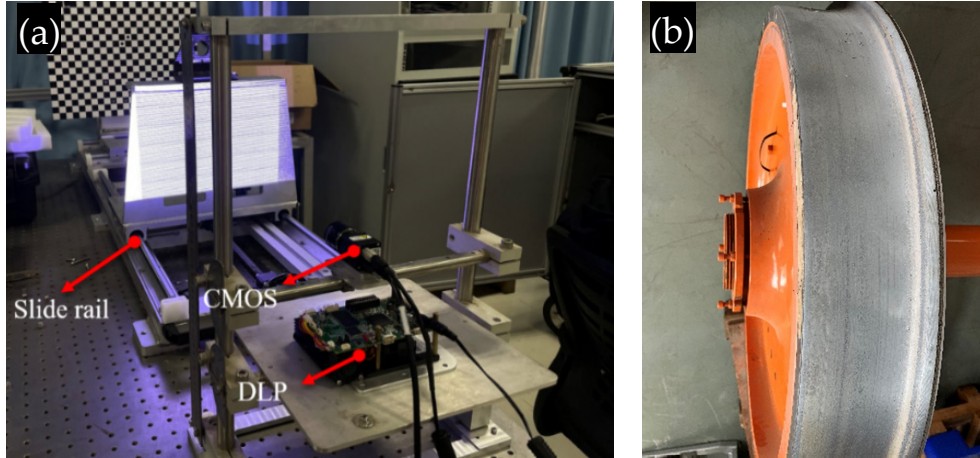

**Figure 2.** (**a**) Experimental system; (**b**) the measured wheel tread object.

## 4. Discussion

The feasibility of the improved Stoilov algorithm was verified by measurement of the wheel tread object shown in Figure 3a. The wheel tread was fixed on the working table, and the five-frame phase-shifted fringe pattern was projected onto its surface by the DLP. Figure 3b shows the single deformed fringe pattern captured by the CMOS camera. The unwrapped phase using the traditional Stoilov algorithm and the improved Stoilov algorithm was established (Figure 4a,c, respectively). The 3D shape reconstruction using the traditional Stoilov algorithm and the improved Stoilov algorithm was also established (Figure 4b,d, respectively). Figure 4b shows the 3D shape of the wheel tread restored by the traditional Stoilov algorithm, where it can be seen that the wheel tread had a large area missing and could not be reconstructed. Even if it could be reconstructed in some places, there were burrs and gullies, and its measurement accuracy was relatively low. From the experimental results, we can see that the improved Stoilov algorithm eliminated the influence of four singularities on the experimental results.

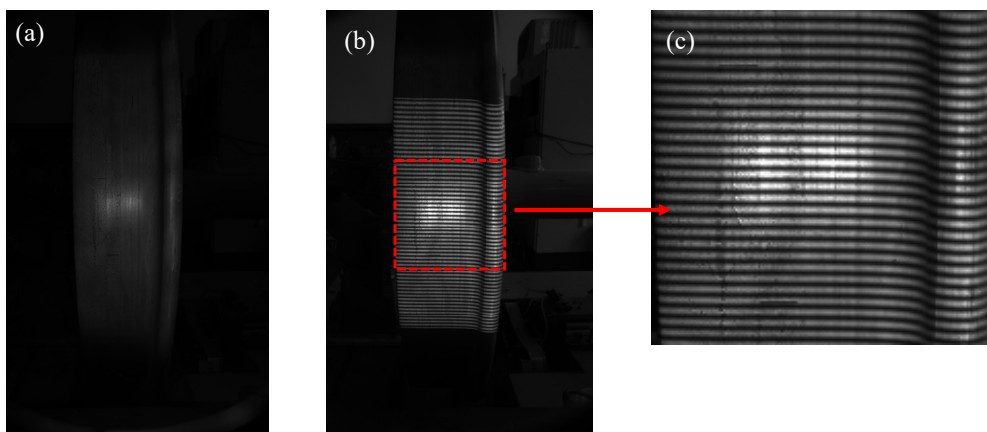

**Figure 3.** (**a**) The measured wheel-tread object; (**b**) deformed fringe; (**c**) enlarged details of the indicated region in (**b**).

To estimate the accuracy of the improved Stoilov algorithm, we measured a flat plane with a height of 5 mm from the reference plane. The height reconstruction measured by the traditional Stoilov algorithm and the improved Stoilov algorithm is shown in Figure 5. It can be seen in Figure 5a that the plane had a large area missing and could not be reconstructed. Although it could be reconstructed in some places, there were burrs and gullies. We can see the reconstruction of the flat plane restored by the improved Stoilov algorithm without missing regions or burrs in Figure 5b. From the results, we can see that

the improved Stoilov algorithm had a better performance in flat plane reconstruction. In the measurement of 3D reconstruction, the average absolute error $\delta = \frac{\sum |h\,(i,j) - A|}{m}$, the average relative error $s = \frac{[|h\,(i,j) - A|/A] \times 100\%}{m}$, and the mean square error $\varepsilon = \sqrt{\sum (h\,(i,j) - A)^2 / m}$ were used to evaluate the accuracy of measurement, where $h\,(i,j)$ is the measured height of the plane, A is the standard height of the plane, and m is the total pixel of the plane. We measured several planes of different heights, and the reconstruction error analysis is shown in Table 1. The measuring system consisted of a projector and a camera. The measuring range was limited, and the measuring accuracy was related to the distance from the measured object to the projection center. When the measured object was just on the focal length of the projector, the measurement accuracy of the system was the highest. Taking the best measuring point as the center in the measurement range, its accuracy was attenuated to both sides. From the data in the Table 1, it can be seen that the best measuring point was about A = 25 mm, with A = 25 mm as the center, and the measuring accuracy gradually attenuated to both sides.

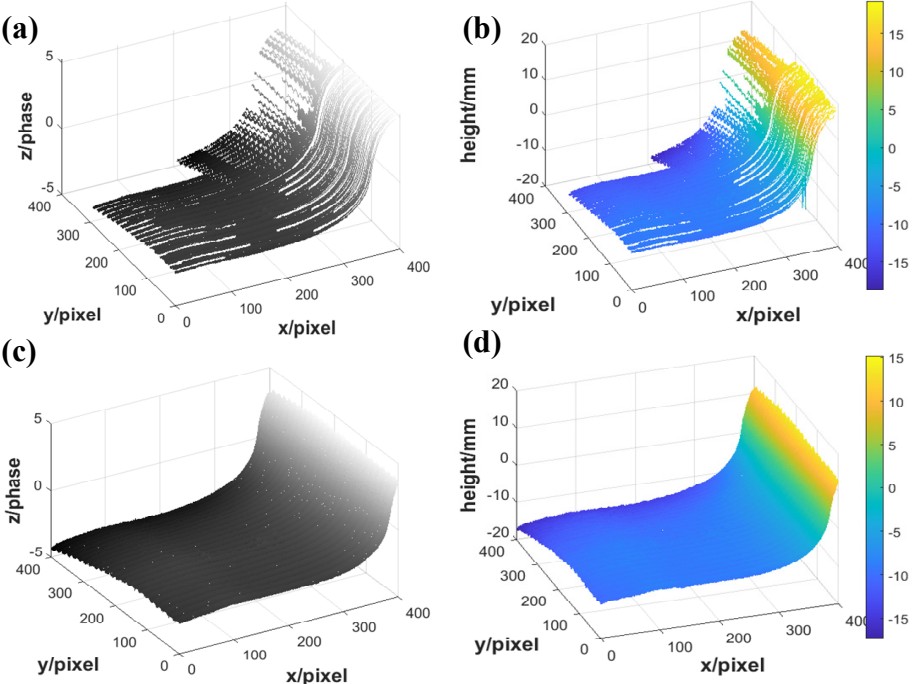

**Figure 4.** Unwrapped phase and reconstruction of (**c**) in Figure 3: (**a**,**b**) using traditional Stoilov algorithm; (**c**,**d**) using improved Stoilov algorithm.

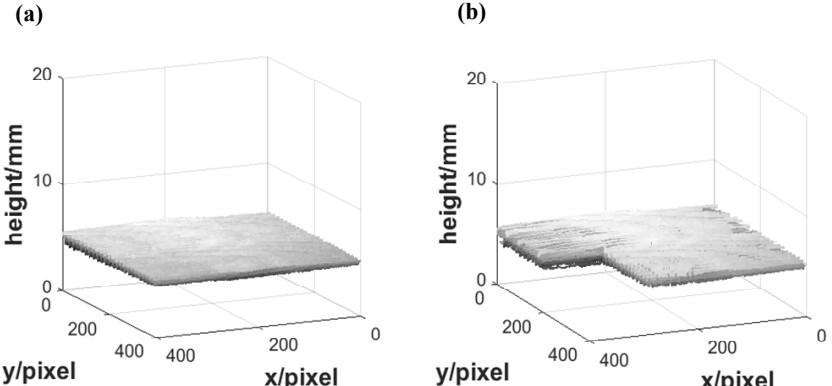

**Figure 5.** The height reconstruction of the plane: (**a**) using traditional Stoilov algorithm; (**b**) using improved Stoilov algorithm.

**Table 1.** Measurement errors of different height planes.

| Category | Measurement Errors of Different Height Planes | | | | | | |
|---|---|---|---|---|---|---|---|
| *A* (mm) | 5 | 10 | 15 | 20 | 25 | 30 | 35 |
| $\delta$ (mm) | 0.2735 | 0.4539 | 0.4277 | 0.2600 | 0.1651 | 0.2806 | 0.3670 |
| *s* | 5.47% | 4.54% | 2.85% | 1.30% | 0.66% | 0.94% | 1.05% |
| $\varepsilon$ (mm) | 0.3393 | 0.5135 | 0.5011 | 0.3338 | 0.2072 | 0.3426 | 0.4435 |

Furthermore, in order to verify the ability of this method to detect intrusions on the wheel tread, we placed a block on the wheel tread and measured it. Figure 6a–d show the acquired deformed fringe pattern, the retrieved unwrapped phase using the improved Stoilov algorithm, the 3D reconstruction of the wheel tread attached with a block, and the height data of column 256 in Figure 6c, respectively. According to the experimental results, the improved Stoilov algorithm had a better performance in the reconstruction, and its repeatable accuracy could be guaranteed; hence, this method can be used to detect intrusions on the wheel tread.

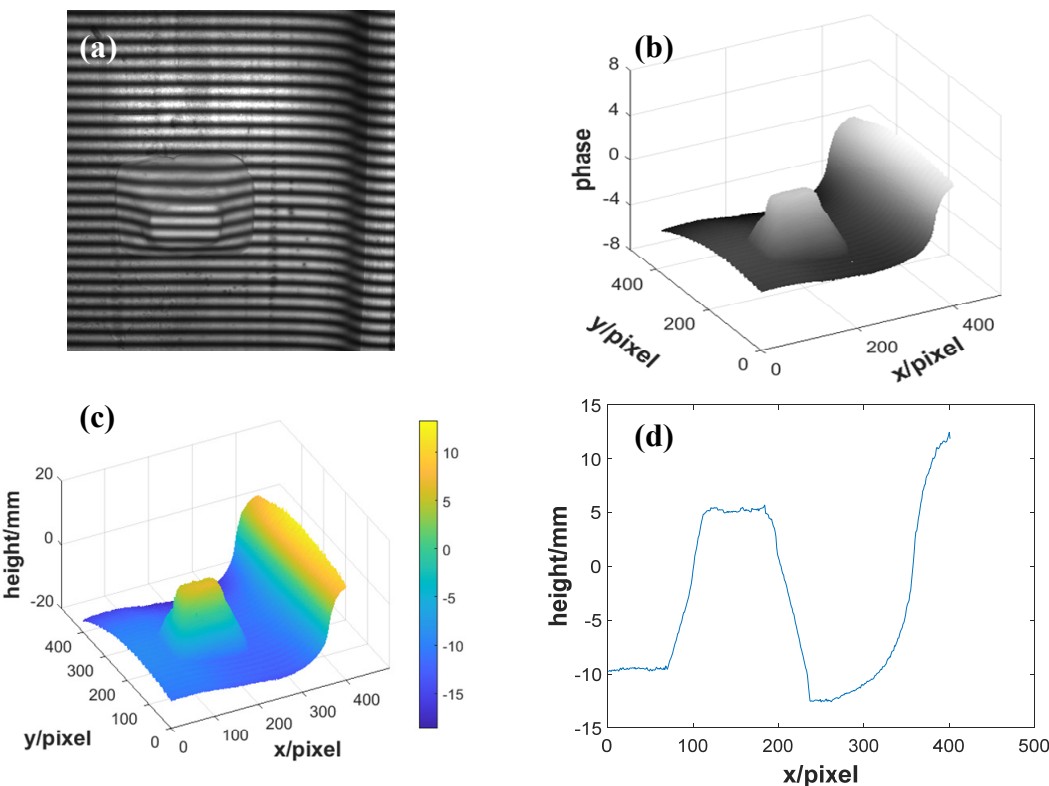

**Figure 6.** Wheel tread attached with a block: (**a**) deformed fringe pattern; (**b**) unwrapped phase by improved Stoilov algorithm; (**c**) 3D reconstruction of the wheel tread attached with a block; (**d**) height data of column 256 in (**c**).

## 5. Conclusions

In this paper, we presented an improved Stoilov algorithm for wheel tread detection. Using this method, the phase error brought about by the abnormal phenomena was restrained, and the precision of 3D measurement was improved. The experimental results showed its feasibility and practicability for 3D shape reconstruction.

The experimental results showed that, compared to the traditional Stoilov algorithm, the improved Stoilov algorithm had better performance in unwrapping the phase and in wheel tread reconstruction. Moreover, to estimate the accuracy of the improved Stoilov algorithm, the plane 5 mm from the reference plane was measured, and the average absolute

error of the reconstructed plane was 0.2735 mm, the average relative error was 5.47%, and the mean square error was 0.3393 mm. Furthermore, in order to verify the ability of this method to detect intrusions on the wheel tread, we measured the tread of a wheel with an intrusion. The experimental results showed the feasibility and the validity of this method in 3D reconstruction of the wheel tread and intrusions on the wheel tread.

In future work, the PMP will be demonstrated for the 3D shape measurement of static objects, and the consistency of the object position is a prerequisite to ensure the successful application of PMP in moving objects. The quantity of phase-shifting steps is not necessarily controlled strictly in Stoilov's algorithm; hence, the equivalent phase-shifting step can be obtained by the movement of the measured object, which is suitable for online phase measurement profilometry. In the future, we intend to project a single-frame sinusoidal stripe pattern onto the rotating wheel tread to further reconstruct the wheel tread profile, so as to realize the dynamic reconstruction of the wheel tread, as well as monitor the safety of the wheel tread profile in real time.

**Author Contributions:** Conceptualization, T.T. and J.P.; methodology, T.T. and J.L.; software, T.T. and R.M.; validation, J.P., J.L. and Y.W.; formal analysis, X.L.; investigation, T.T. and J.P.; resources, J.P.; data curation, T.T.; writing—original draft preparation, T.T.; writing—review and editing, Y.W.; visualization, T.T. and Y.W.; supervision, J.P. and J.L.; project administration, J.P. and J.L.; funding acquisition, J.P. All authors have read and agreed to the published version of the manuscript.

**Funding:** This research was funded by the Natural Foundation International Cooperation Project, the funding number is 61960206010.

**Institutional Review Board Statement:** Not applicable.

**Informed Consent Statement:** Not applicable.

**Data Availability Statement:** Not applicable.

**Acknowledgments:** The authors are grateful to the Institute of Optoelectronic Engineering, School of Physics, Southwest Jiaotong University for administrative support and equipment.

**Conflicts of Interest:** The authors declare no conflict of interest.

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
