# Peer review of "Wheel Tread Reconstruction Based on Improved Stoilov Algorithm"

_optics, doi:10.3390/opt3020016_

Round 1

Reviewer 1 Report

Reviewer’s comments on manuscript „Wheel tread reconstruction based on improved Stoilov algorithm” by T. Tang et al.

This manuscript, dealing with fault detection in train wheel with fringe/phase profilometry, is both interesting and practically important.      

Subjectal concerns:

  1. On Page 1, after introducing an abbreviation like PMP, please give the full definition right after the first occurence.
  2. Regarding Fig. 1, on Page 2, illustrating the PMP method, much more verbal explanation would be needed in the text: please explain the meaning of xr, x0, d, L, and the meaning of the lines symbolizing light rays. Where are used the xr, x0, d, L parameters?
  3. The connection between F(x,y) and y(x,y) is mentioned on Page 2. Please state that connection explicitely as a mathematical formula. Please right the same y(x,y) in both the text and formula (both normal or upper case).
  4. On Page 3, when describing „Improved Stoilov algorithm” the verbally summarized (1-5) points, which also constitute the essence of this paper: Can it be extended with a more formal statistical treatment with the language of Maximum Likelihood (ML) estimation? In this case f0 or sin(f0) could be obtained more formally, by maximising – or finding the place of maximum of – an appropriate likelihood function. If it can be done, involving it in the text is suggested. Alternatively, the difference of the two should be mentioned.
  5. On Fig. 2, on Page 3, where is the wheel with the tread? It can not be seen. Please give another photo and/or indicate it with an arrow. Perhaps the most useful would be giving another – inset – photo only of the wheel tread, magnified.       
  6. On the wheel heads of Fig. 3, on Page 4, please indicate the deformed regions by „encircling” and arrows. The deformations are at most hardly seen, please enlarge those portions of the images.
  7. Please designate on Fig. 3 which portions of the wheel do correspond to the regions of the surface plots of Fig. 4. Or please refer to these portions/regions in Fig. 3 in the legend of Fig. 4.
  8. According to Table 1, on Page 5, the absolute and relative errors are minimal with A= 25 mm, and they seem to be periodic with size „A”. Can this be true? What is the explanation? It should be discussed in the text.                    

Formal concerns:

  1. Sometimes the spaces are omitted in the text, e.g. after right parentheses „]” and after dots „.”. This can occur e.g. after reloading a text in „MDPI” format, perhaps depending on software version.
  2. On Page 3, Line 9, right parantheses „)” should be of normal size, instead of subscript.
  3. On Page 6, Line 17, please revert „in that” to the correct „that in” .
  4. Please restructure the text into: Introduction, „Materials and methods”, „Results”, „Discussion” (or „Results and discussion”), and to „Conclusion”.

Author Response

Dear Prof. Hogan Zhang and Reviewers:

Thank you for your letter and for the reviewers’ comments concerning our manuscript entitled “Wheel tread reconstruction based on improved Stoilov algorithm”. (Manuscript ID: optics-1637961). Those comments are all valuable and very helpful for revising and improving our manuscript, as well as the important guiding significance to our researches. Thank you very much for reminding me that the similarity of the manuscript is 43% (without affiliations and references), and we have revised my manuscript according to the similarity reports. We have studied comments carefully and have made correction which we hope meet with approval. Revised portion are marked in red in the manuscript. The main corrections in the manuscript and the Response to the reviewer’s comments are as flowing, the revised portion are marked in red in this response, and highlighted with “track changes” in the manuscript.

Reviewer 2 Report

Improved Stoilov algorithm and the unwrapped phase is applied to make three-dimensional profile of the wheel tread so authors could get the closest value to true phase shift using the method of probability and statistics. However, authors need to show advantages and disadvantages of the previous research in detail.

  1. Please describe advantages and disadvantages of the previous research in detail from Introduction section.
  2. Authors should use abbreviated journal names in the reference section.
  3. Please make space 2] and by. Please check other references in entire manuscript.
  4. In Equation (1), why maximum of n is 5 ?
  5. Please use Equation instead of Eq. because of MDPI styles.
  6. In Figures 4 and 5 , size of x- and y-labels are small to be seen.
  7. Figure 6 label sizes and values are small.
  8. In conclusion section, please show some important value data.
  9. Please show some future work in conclusion section.
  10. Between page 7 and page 8, there are big empty space.
  11. Please show manufacture information of the experiment system such as Light-Crafter 4500, Basler acA1920-40gm, and GW12H4R1200ZFCM.

Author Response

(The authors gave the same response as above.)

Round 2

Reviewer 1 Report

The authors carefully answered all my questions and carried out all suggested alterations on the manuscript, which I accept. From my part the manuscript might be accepted for publication.